# The Peer Effect in Promoting Physical Activity among Adolescents: Evidence from the China Education Panel Survey

**DOI:** 10.3390/ijerph20032480

**Published:** 2023-01-30

**Authors:** Zhengqing Zhou, Xinchen Li, Zhanjia Zhang

**Affiliations:** Department of Physical Education, Peking University, No. 5 Yiheyuan Road, Haidian District, Beijing 100871, China

**Keywords:** peer effect, physical activity, junior high school students

## Abstract

For a long time, studies on the peer effect of physical activity among adolescents have focused on relevance rather than causality. This article provides empirical evidence of the peer effect of physical activity among adolescents using data from the China Education Panel Survey. The results show that the peer effect increases physical activity by about 6.757–8.984 min per week among classmates, a finding consistent with previous studies. Using the instrumental variable approach and considering the potential missing variables, the peer effect increases physical activity by 23.923–27.410 min per week, representing a threefold increase. In addition, the general attitude towards sports in class plays a significantly influential role, accounting for 20% of the peer effect of physical activity.

## 1. Introduction

The Healthy China Action (2019–2030) explicitly proposes implementing health-promotion activities for primary and secondary school students “to ensure sufficient sport activities and cultivate lifelong sport habits” as an important goal (for brief explanations on the Healthy China Action plan, see http://www.chinadaily.com.cn/a/201907/17/WS5d2ecd4ba310d830563ff8ba.html (accessed on 29 January 2023)). In recent years, a large number of studies have been published on how peers promote physical activity in adolescents and the heterogeneity of such promotion among different genders and ethnic groups. More specifically, many scholars attest that good peer relationships [1], a common social environment [2,3], peer company [4], the time of company [5], information guidance [6], and peer support [7] have a positive role in promoting physical activities among adolescents.

Although Xia Shuaibing and Liu Shujun [8] urged scholars to pay attention to the recognition of the causality of peer effects in sports research, published studies have largely been limited to correlation analysis, lacking causality inference. Hence, the conclusions and policy recommendations derived from such research should be treated with caution. Recognizing the causality of peer effects on individuals’ physical activity requires focusing on the problem of the endogeneity of peer groups. First, similar behaviors of peers result from mutual selection by individuals with similar values, preferences, or characteristics [9]. In other words, the peer effect on individuals’ behaviors may involve interactions of both cause and effect rather than one-way causality. Second, regardless of one-way causality or mutual selection, the extent to which a peer group plays a role depends on the individual’s opportunity to establish a relationship with peers; that is, there is an issue of possible missing variables [10]. For instance, the structural characteristics of a school or residential community can affect the closeness and opportunities for meetings with peers. Therefore, establishing a peer relationship is not only based on the individual’s preferences but also constrained by existing opportunities.

According to Manski [9,11], the endogeneity is mainly manifested by the inverse causality and missing variables in this area. However, Ryan [12] suggests that the endogeneity of the peer effect can be identified by finding exogenous variables. The main objective of this article is to examine whether the peer effect constitutes a causal relationship with physical activity among adolescents, and discuss the potential channels that exist to detect the such effect. It verified the hypothesis proposed by Wang Fu Bai Hui et al. [6], Quan Xiaojuan, and Lu Chuntian [13] that peer support has a significant positive effect on an individual’s time spent exercising. Additionally, analysis of the mechanism showed that the atmosphere of sports in a class plays a mediating role, as the peer effect influences the physical activity time among adolescents, and the peer effect increases the physical activity time by promoting the atmosphere of sports in a class.

## 2. Literature Review

### 2.1. The Concept of the Peer Effect

In 1966, the peer effect was applied to the study of education for the first time in the famous Coleman Report [14], which held that the peer effect was an important factor affecting the academic achievement of students and had a great impact on the educational policy of the United States. Later, the peer effect was extrapolated to perform research on more social behaviors, including learning [15,16,17]; sports [4,5]; and adolescent misconduct, such as substance abuse [18]. The peer effect defined in this article refers to a broad concept that encompasses the positive effect of peer behavior and the negative effect of “destructive” students [19]. It comes from a peer group of similar ages, with similar hobbies, life attitudes, and goals [20], rather than from the organizational group in the narrow sense [21].

Why are peer groups so important for promoting adolescent sports? As children reach puberty (about 10 to 19 years of age), they gradually shift their attention from their parents to their peers; are willing to spend more time with their peers; and the influence of the peer group transcends that of parents, the family, and the school as an important factor in raising the level of physical activity of adolescents [13,22].

The premise of the peer effect is that people exist within a group [15]. Classmates or schoolmates, children of family friends and relatives, children living in the same community, and friends from social networks are the main sources of peers for junior high school students. As for the growth environment of junior high school students, their classroom is an important place for them to learn and spend time in, and the behavior and output of peers in their class have an important influence on the behavior or output of other students of the same class [15]. The peer effect in the class has been verified extensively in education [17,23,24], but its influence on an individual’s physical activity in the class has seldom been investigated. Intervention research on adolescent physical activity has been carried out based on the social network of the peers of the class but has yielded inconsistent conclusions [25].

### 2.2. Identifying Problems Is Difficult

In sports and other social sciences, recognizing the causality of peer effects is a primary challenge. Manski proposed three interactions between an individual and his peer group [9]: (I) There are endogenous effects; i.e., the individual’s behavior is affected by group behavior. For instance, a keen class atmosphere of sports naturally causes students to actively engage in physical exercise. (II) There are also contextual effects; i.e., the individual’s behavior tends to be influenced by the external characteristics of the group. For instance, the individual student’s enthusiasm for exercise is influenced by whether there are more exercise-loving children in the class or whether parents of other students are willing to urge their children to exercise. (III) Lastly, there are correlated effects; i.e., the behavior of individuals in the same group is similar because of the influence of common environmental characteristics or institutional arrangements, such as a capable physical education teacher, a learning schedule, and investments in sports facilities.

## 3. Methodology

### 3.1. Study Design

This study used data from the China Education Panel Survey (CEPS), which used the 2013–2014 school year as the baseline and two concurrent cohorts of the seventh (first year) and the ninth (third year) grades of junior high school as the starting point. The CEPS applied a stratified, multistage sampling design with probability proportional to size (PPS), randomly selecting a school-based, nationally representative sample of approximately baseline students in 438 classrooms of 112 schools in twenty-eight county-level units in mainland China (for sampling design, see http://ceps.ruc.edu.cn/English/Overview/Overview.htm (accessed on 29 January 2023)). All the students in the selected classes were sampled. This survey provided good data support for the study. In the 2014–2015 school year, 9449 seventh-grade students enrolled in the baseline survey were followed up successfully.

To reduce interactions as both cause and effect from the interactions between the self and the peer group in the peer-effect study, we considered that the possibility of interactions was low among the students who had just entered the class at the baseline survey and that the peer effect could be observed among classmates who had remained in the same class for more than one year at the follow-up survey. Following the practice of Quan Xiaojuan and Lu Chuntian [13], we only analyzed a total of 7843 students who had remained in the same classes between the follow-up and baseline surveys.

### 3.2. Variables and Measures

Specifically, the primary variables were processed as follows:(1)Explained variable. The explained variable of this article is the individual exercise time of junior high school students. The CEPS surveyed the time of physical activity, i.e., the number of days of physical activity per week and the time of physical activity per day (minutes) of the seventh-grade junior high school students in the school year of 2014–2015. The authors calculated the two survey items in terms of their respective weekly cumulative time. To reduce the impact of outliers on the estimates, the authors used the STATA winsor2 command for winsorization and replaced the case values over 300 min per week with 300 min.(2)Core explanatory variable. The core explanatory variable of this article is peer behavior. In society, individual and peer behaviors often occur simultaneously, interacting with each other as cause and effect. The use of peer characteristics at baseline as a surrogate measure of peer effect is a commonly used approach to address this issue. According to Zhao Ying [24], the average time spent on extracurricular sports with other people from Monday to Friday at the baseline survey was regarded as the measure of peer effect in the class.(3)Control variables. According to the literature, the control variables in this article are individual characteristics, including gender, age, registered permanent residence, academic level, boarding at school or not, being an only child or not; family characteristics, including the education level of parents and family economic status; class characteristics, including the proportion of students residing in rural areas, the ranking of the class’ academic performance in the school; and school sports facilities, including circular runways, sports grounds, gymnasiums, and swimming pools.

### 3.3. Method Derived

The mainstream method of estimating the peer effect involves using the linear mean model [10]. On the one hand, the model is structurally indirect and has strong explanatory power and good correspondence to the different peer effects proposed by Manski (1993, 2000). On the other hand, the model has strong expandability and can further construct nonlinear peer effects, as well as determine the difference between the individual and the influence of different groups of peers on the individual *i*. Ryan (2017) set the benchmark model as the following equation [12]: (1)$$A_{igst}=\theta_{1}+X_{igst}^{\prime }\theta_{2}+{\bar{P}}_{-igst}^{\prime }\theta_{3}+\varphi_{s}+\lambda_{t}+\varepsilon_{igst}$$
where $A_{igst}$ is the explained variable of the individual *i* of the class *g* of the school *s* at the period *t*, and ${\bar{P}}_{-igst}$ is the average observable characteristic of the peers other than *i*, excluding other factors in the individual’s observable characteristic vector $X_{igst}$. In addition, school $\varphi_{s}$ and time $\lambda_{t}$ are used to represent the fixed effects of school and time, respectively, and $\varepsilon_{igst}$ representing the error term is used to capture the non-observed random influencing factor. If the student’s gender influences $A_{igst}$, it appears both as an individual characteristic in the vector $X_{igst}$ and as a peer characteristic in ${\bar{P}}_{-igst}$.

Ryan (2017) summarized the five main variants of Equation (1) according to the literature [12], and for each equation, $\theta_{3}$ is the peer effect of interest. The first variant is ${\bar{P}}_{-igst}={\bar{X}}_{-igst}^{*}$, and the latter uses a subset of individual characteristics as peer characteristics. This method estimates contextual effects and still needs to solve the problems of reverse causality and missing variables. The second variant is ${\bar{P}}_{-igst}={\bar{A}}_{-igst}$, and the latter is the “leave-one-out mean” of the explained variable at the same period; this method estimates the endogenous peer effect [26]. Particularly, under this framework, it is necessary to add the instrumental variable ${\bar{Z}}_{-igst}^{*}$ and the individual characteristic ${\bar{X}}_{igst}$ to estimate the mean peer effect of the equation interpretation, such as Equation (2): (2)$$\begin{array}{cc} {\bar{A}}_{-igst}^{*}= & \delta_{1}+X_{igst}^{\prime }\delta_{2}+{\bar{Z}}_{-igst}^{*}\delta_{3}+\kappa_{s}+\tau_{t}+\xi_{igst} \end{array}$$

The disadvantage of this approach is that it is impossible to distinguish between the endogenous and the contextual effects because *X* directly (in Equation (2)) or, via ${\bar{A}}_{-igst}^{*}$, indirectly influences the explained variable $A_{-igst}^{*}$.

The third variant is ${\bar{P}}_{-igst}=\{{\bar{A}}_{-igst},{\bar{X}}_{-igst}^{*}$}, which supplements the characteristic of the peer population $\bar{X}$ based on Equation (2). The fourth variant is ${\bar{P}}_{-igst}={\bar{Z}}_{-igst}^{*}$, which is essentially the classical instrumental variable method and is more exogenous than the instrumental variable in Method 2 [27,28]. The last variant is ${\bar{P}}_{-igst}={\bar{A}}_{-igst-k}$, in which the peer characteristics of the t−k period logically may not influence the behavior of the individual at the current period, thus avoiding the problem of reverse-causality to some extent. Nevertheless, some characteristics of the peers may still be related to the individual at the current period over time, and there are still some problems relating to influence (Manski, 1993). Hanushek et al. [16] held that frequently occurring common shocks cause a positive correlation, often leading to a downward bias in the estimate; hence, the estimate can be considered as the lower limit of the endogenous peer effect.

In response to criticism from Angrist (2014) [29], Ryan held that the model was not designed to address the issue of causal inference [12]. The data-generation process cannot be controlled as precisely in social sciences as in natural sciences, and the investigator can only analyze data as an observer. Hence, to solve the problem of endogeneity fundamentally, additional assumptions must be applied to identify them. The ideal approach is to conduct random experiments (see Duflo [30] and the well-known STAR program) in addition to quasi-experimental studies such as fixed-effect models, instrumental variables, propensity scores, and breakpoint regression [15].

### 3.4. Analytical Strategy

Analyses were conducted with the statistical software package STATA, version 17.0. Descriptive statistics including sample size, means, standard errors and value range were calculated for all study variables. As the analysis above, this article still uses the conventional linear mean model, extrapolates Method IV of Ryan (2017), and sets the benchmark model as Equation (3) to verify the impact of the peer effect on junior high school students’ physical activity in the class [12].
(3)$$PA_{igst}=\beta_{1}+X_{igst}^{\prime }\beta_{2}+{\bar{PA}}_{-igst-1}^{\prime }\beta_{3}+\varphi_{s}+\varepsilon_{igst}$$

In terms of variables, the explained variable $PA_{igst}$ represents the level of physical activity of the student *i* in the class (cohort or group) *g* of the school *s* in the year *t*; ${\bar{PA}}_{-igst-1}$ represents the level of exercise of peers other than the student *i* at baseline; $X_{igst}$ is the individual’s external characteristics of student *i* during the same period, including those at the individual, family, class, and school levels; and $\varepsilon_{igst}$ is the error term. In terms of parameters, $\beta_{3}$ is the peer effect of the primary identified object. The analysis strategy of this article is as follows:

First, correlation coefficients were obtained but may be biased. The individual control variable, class control variable, school control variable, and county/district fixed effect are gradually introduced into Equation (3) to examine the influence of the core explanatory variable of peer effect on adolescents’ physical activity.

Second, the causality coefficient was obtained and compared with the former. In addition to the core explanatory variable, many articles [7,31] report other factors that influence physical activity. The relationships of these factors with peer behavior and the individual’s physical activity make it impossible to exclude bias in the estimates. Therefore, we used a two-stage least squares (TSLS) estimation method of instrumental variables, which is required to address the above problem of endogeneity of peer effects to capture their causality. Based on the model setting of Equation (3), the first-stage regression estimation equation for the individual exercise participation is as follows: (4)$$\begin{array}{c} {\bar{PA}}_{igst}=\rho_{1}+X_{igst}^{\prime }\rho_{2}+{\bar{Z}}_{-igst-1}^{\prime }\rho_{3}+\varphi_{s}+\xi_{igst} \end{array}$$
where ${\bar{Z}}_{-igst-1}$ is the instrumental variable of the mean frequency of parent-accompanied exercise of peers at baseline, and the other variables are the same as those in Equation (3). In this article, the two-stage least squares (TSLS) method is used to estimate the peer effect $\beta_{3}$. In the first stage, the exercise behavior of peers in the current period is estimated, and the fitted value of ${\bar{PA}}_{igst}$ is obtained via Equation (4), and in the second stage, it is substituted into Equation (3) to obtain the TSLS estimate of $\beta_{3}$, i.e., ${\hat{\beta }}_{3}^{TSLS}$.

Finally, because empirical evidence of the mechanism of peer effects is inadequate, the intermediate variable of the sports atmosphere is introduced in this article to further understand its influencing channel.

## 4. Results

### 4.1. Benchmark Regression

Descriptive statistics for all variables are presented in Table 1.

Table 2 displays the OLS regression results reflecting the influence of the average weekly exercise time of other students on the exercise time of a student in junior high school classes. The explained variable from columns (1) to (5) is the individual exercise time (minutes/week). First of all, according to the regression results of column (1), it was found that the individual exercise time increases by 8.984 min per day for the average increase of one standardized unit of the class peers. On this basis, columns (2) to (4) further add the individual characteristic variables (gender, rural resident or not, only child or not, and academic level), class characteristic variables (ratio of rural residents in the class and ranking of the class’ academic performance in the school), and the school’s sports facilities (with or without circular runways, swimming pools, gymnasiums, and stadiums). Overall, the peers’ exercise time shows a significant positive effect on the individual exercise time at the 1% level in all models. As shown in column (5), after controlling the fixed effect of county and district, the influence on the result is still small, with the peer effect of exercise time slightly reduced to about 7.343 min per day. The above empirical results show the robustness of the conclusion derived from the models, and the hypothesis that the peer effect of exercise is positive is thus verified.

### 4.2. Instrumental Variables Regression

Theoretically, to satisfy the strong correlation and exogeneity, an instrumental variable should first be highly correlated with peer exercise participation at baseline, and second, it cannot affect the individual’s exercise at the current period directly—only indirectly, by affecting peer exercise participation at baseline. Specifically, according to Jin Honghao et al. [32], the related characteristics of peers’ parents can serve as the instrumental variable of peer behavior. Based on this, we used the mean frequency of parent-accompanied peer exercise at baseline as the instrumental variable to solve the endogeneity problem in this article. The rationale is as follows: given exogeneity, there is no obvious correlation between the frequency of parent-accompanied exercise of peers and other factors that may affect the individual’s participation in exercise; hence, the exogeneity is satisfied. In terms of correlation, parent-accompanied exercise is closely related to the formation and development of children’s exercise habits, thus meeting the requirements of the direct correlation between the instrumental and the endogenous explanatory variables. Therefore, in this section, the mean frequency of parent-accompanied exercise of peers at baseline serves as an instrumental variable to address the possible endogeneity problem in peer-effect analyses.

Specific results are shown in Table 3. The regression results in column (2) show that, after considering the problem of endogeneity, the peer exercise time still has a significant positive effect on the individual’s sports participation; an increase in the peer exercise time by one standardized unit increases the individual’s weekly exercise time by nearly 23.923 min, and there is still a significant positive effect at the level of 1%. In one-stage regression, the regression coefficient of the instrumental variable was approximately 0.48 and significant at the level of 1%. This indicates a positive correlation between parent-accompanied exercise and peer exercise, which is consistent with the conclusion of Jin Honghao et al. (2021). Of course, the premise of the TSLS results is that the mean frequency of parent-accompanied exercise of peers at baseline is an effective instrumental variable (There are many reasons for identifying problems with the first-stage regression; for example, weak tool variables can cause problems that cannot be identified). For this reason, a weak instrumental variable test should be performed first. The F statistic of the first-stage regression result of the instrumental variable in column (2) is 23.598 (far greater than the empirical value of 10), which refutes the original hypothesis that the first-stage regression poses an identification problem. The result of the Anderson–Rubin test also led to the rejection of the hypothesis of a weak instrumental variable (the *p*-value is about 0.001) (Considering that LIML estimates are generally affected less by weak instrumental variables, we also calculated the limited information maximum likelihood estimate (LIML). However, we found no significant difference between the LIML estimate and the TSLS estimate). The premise of using the instrumental variable method is the existence of an endogenous explanatory variable, and the robustness DWH-test statistic is about 17.31 (*p*-value < 0.0001), strongly rejecting the original hypothesis that the peer exercise time is exogenous, which confirms the necessity of using the instrumental variable method for estimation (Under the heteroscedasticity condition, the DWH test is more robust, and its idea is derived from the Hausman test. The Hausman test determines whether an explanatory variable is endogenous. IV is used only if the peer effect is endogenous; if the peer effect is not endogenous, IV is not required. If the regression results are quite different, then at least one of all explanatory variables (including peer effects) is endogenous and, therefore, IV is necessary).

The regression results in column (3) show the robustness of the test results with the addition of another instrumental variable (the abundance of individual-participated activity). The results show that the influence coefficient of peer exercise time on an individual’s sport participation is 24.290, indicating an increase of about 2%, which means that the increase in peer exercise time by one standardized unit increases the individual exercise time by 24.290 min per week, and it is significant at the level of 1%. Except for the core explanatory variable, there is little change in the t value of the parameter estimates for the other control variables. This reflects the robustness of the original instrumental variable to a certain degree. The optimal GMM estimation model parameters were used as a surrogate for the estimation method, and the peer effect of exercise increased to 27.410 min. Because the number of instrumental variables was greater than that of endogenous variables, the exogeneity of instrumental variables could be tested. The results show that the *p*-value is 0.575, greater than the preset value of 0.05. Therefore, the original explanation cannot be rejected; that is, the overidentification constraint is valid.

### 4.3. Analysis of Mechanism

Besides the direct causality of the peer effect on the time of an individual’s sport participation, the authors also investigated the mechanism of peer effects on junior high school students’ exercise. According to the literature review above, the peer effect plays an important role in the formation of an individual’s sports interest (Carnes and Barkley [33]), and the individual’s interest, enjoyment, and pleasure in sports itself have a significant influence on their sport behavior (Chen et al. [34]; Bao Ran et al. [7]). That is, the peer effect affects the exercise behavior of junior high school students by promoting their interest in sports. Based on the above analysis, we proposed the hypothesis that an interest in and hobby of sports play a mediating role in the influence of the peer effect on an individual’s exercise. To verify this hypothesis, based on Equation (3), the above path model was sequentially tested, and the mediation model was constructed as follows: (5)$${\bar{Interest}}_{igst}=\gamma_{1}+X_{igst}^{\prime }\gamma_{2}+{\hat{{\bar{PA}}^{\prime }}}_{-igst-1}a+\varphi_{s}+e_{igst}$$
(6)$$PA_{igst}=\gamma_{1}+X_{igst}^{\prime }\gamma_{2}+{\hat{{\bar{PA}}^{\prime }}}_{-igst-1}c^{\prime }+{\bar{Interest}}_{igst}b+\varphi_{s}+\varepsilon_{igst}$$

In Equation (6), the explained variable is the individual exercise time, and the explanatory variable is the peers’ exercise time. Based on Equation (3), the proportion of students who are interested in sports in the class of which the individual is a member during the current period (in this article, this proportion is used as the measure of the atmosphere of sports in a class), and ${\bar{Interest}}_{igst}$ is added to measure the intermediary effect from the interest in sports. The explained variable in Equation (5) is the current atmosphere of sports, which is used to investigate the influence of the peer effect on forming a current interest in sports. Because the peer effect of exercise has the problem of endogeneity, the peer-effect-fitted value of the first-stage regression ${\hat{{\bar{PA}}^{\prime }}}_{-igst-1}$ is used as the explanatory variable, and the test results are shown in Table 4.

The IV1 column of Table 4 shows that the total effect of the fitted value of peers’ exercise time obtained with the mean frequency of parent-accompanied exercise of peers at baseline as the only instrumental variable on individual exercise time is 23.888 (slightly lower than the TSLS_1 result in Table 3, i.e., 23.923), which equals the direct effect of 13.892 plus the indirect effect of 4.936. In addition, the Sobel test value is very small (*p*-value < 0.005), suggesting that the atmosphere of sports in a class as a peer effect is statistically reliable as an intermediate variable of junior high school students’ sports participation. In addition, the peer effect is reduced by approximately 20.7% after eliminating the intermediary effect. Similarly, we controlled the fitted values of two instrumental variables on the peer effect in column IV2 and found that the total effect of peer exercise time on an individual’s sports participation and the intermediary effect of the atmosphere of sports in a class show robustness, and the intermediary effect accounts for about 19.5% of the total effect.

## 5. Discussion and Conclusions

Our findings suggest that the peer effect is causally related to adolescents’ physical activity and discuss potential channels through which such an effect might be detected. We used the peer exercise time at baseline as a surrogate variable for the peer effect and found that it exerts a significant positive effect on an individual’s exercise time. Consistent with previous studies in the field [6,13], the peer effect increases the weekly exercise time of other students in the class by about 6.757–8.984 min, which is statistically significant, although the effect is not large.

The key question is whether this association is causal and whether the peer effect has a greater influence on physical activity than previously thought. Based on the research of Manski [9,11] and Ryan [12], we show that the problem of identifying peer effects of exercise lies mainly in estimation bias due to missing variables. Specifically, a comparison of the TSLS results with the OLS results (column (1) in Table 3) shows that the IV-TSLS-based estimation coefficients of peer exercise time increase to 23.923–27.410 min/wk, is inflated by 3.5–4 times, either by increasing the number of instrumental variables or by using the optimal GMM method, suggesting a downward bias in the OLS estimation. This conclusion is consistent with what Manski [9,11] and Angrist [29] suggested, i.e., omitting the factors positively correlated with peer effects is the main cause of OLS underestimation. If the endogeneity is due to reverse causality or measurement error, the OLS should be higher than the true value. These results are robust to the addition of another instrumental variable (the abundance of individual-participated activity) and other control variables, such as the number of friends, the score of negative friends, the score of positive friends, the relationship with classmates, and cognitive ability, as suggested by Quan Xiaojuan and Lu Chuntian (2020) and Wang Fu Baihui (2018).

Furthermore, to investigate the mechanism of the peer effect on physical activity among junior high school students, we examined the mediating role of the class’s sports atmosphere and found that this factor accounted for nearly 20% of the peer effect on physical activity. This conclusion not only theoretically verifies the importance of the peer effect in promoting sports among adolescents, but also provides practical policy inspiration for guiding students to participate in physical activity. To build a healthy China, our suggestion for policies promoting adolescent exercise involves paying attention to the influence of the peer effect on students’ exercise behavior and using it to cultivate students’ interest in sports.

## Figures and Tables

**Table 1 ijerph-20-02480-t001:** Descriptive Statistics of Variables.

	Sample Size	Mean	SD	Min	Max
Explained variable					
Exercise time (min/wk)	7607	137.371	95.37	0	300
Explanatory variable					
Peers’ exercise time (not standardized)	7532	142.915	39.77	39	261
Control variables of individual characteristics					
Boy or not (=1)	7843	0.520	0.50	0	1
Rural resident or not (=1)	7843	0.480	0.50	0	1
Only child or not (=1)	7745	0.491	0.50	0	1
Academic level	7676	3.062	1.07	1	5
Boarding at school or not (=1)	7843	0.234	0.42	0	1
Control variables of family characteristics					
Parents’ education level	7825	0.684	0.82	0	2
Family’s economic status	7821	1.884	0.48	1	3
Control variables of class characteristics					
Proportion of students residing in rural area	7843	2.278	1.02	1	4
Ranking of class’ academic performance in school	7843	1.913	0.54	1	3
Control variables of school characteristics					
Are there circular runways? (=1)	7843	1.091	0.29	1	2
Are there swimming pools? (=1)	7363	0.066	0.25	0	1
Is there a gymnasium? (=1)	7363	0.277	0.45	0	1
Is there a stadium? (=1)	7526	0.974	0.16	0	1
Instrumental variables					
Mean frequency of parent-accompanied exercise of peers	7823	3.739	0.78	1	5
Baseline parental support level	7431	4.615	2.55	2	12

**Table 2 ijerph-20-02480-t002:** Benchmark Model Regression Results.

Explained Variable	Individual Exercise Time				
	**Benchmark Equation**	**(1) + Individual**	**(2) + Class**	**(3) + School**	**(4) + FE**
	**(1)**	**(2)**	**(3)**	**(4)**	**(5)**
Core explanatory variable					
Peers’ exercise time (standardized)	8.984 ***	8.321 ***	7.403 ***	8.543 ***	7.343 ***
	(8.14)	(7.42)	(6.66)	(7.20)	(2.79)
Control variables at individual level					
Boy or not (=1)	32.415 ***	31.140 ***	31.631 ***	31.065 ***	32.503 ***
	(14.80)	(14.14)	(14.43)	(13.79)	(10.11)
Rural resident or not (=1)	−18.118 ***	−3.845 ***	3.496 ***	3.614 ***	2.405 ***
	(8.16)	(1.50)	(1.32)	(1.34)	(0.91)
Academic level	6.902 ***	6.171 ***	6.289 ***	5.884 ***	5.940 ***
	(6.44)	(5.73)	(5.87)	(5.35)	(4.30)
Sample size	7149	7066	7066	6613	6613
Adjusted $R^{2}$	0.059	0.075	0.086	0.099	0.159
Individual characteristics	No	Yes	Yes	Yes	Yes
Class characteristics	No	No	Yes	Yes	Yes
School characteristics	No	No	No	Yes	Yes
County/district fixed effect	No	No	No	No	Yes

Note: t statistics in brackets, *** *p* < 0.01.

**Table 3 ijerph-20-02480-t003:** Instrumental Variables Regression Results.

Explained Variable	Individual Exercise Time			
	**OLS**	**TSLS_1**	**TSLS_2**	**TSLS_3**
	**(1)**	**(2)**	**(3)**	**(4)**
Core explanatory variable				
Peer exercise time (standardized)	6.757 **	23.923 ***	24.290 ***	27.410 ***
	(2.48)	(3.76)	(4.22)	(3.68)
Control variables at individual level				
Boy or not (=1)	33.280 ***	33.938 ***	33.960 ***	33.405 ***
	(10.43)	(11.52)	(11.55)	(10.98)
Rural resident or not (=1)	1.777	−1.493	−1.459	−13.537 ***
	(0.77)	(0.51)	(0.50)	(2.84)
Peer-support-related control variables				
Number of friends	0.146 *	0.167 **	0.167 **	0.121
	(1.73)	(2.19)	(2.19)	(1.61)
Score of negative friends	2.065 **	2.414 ***	2.417 ***	2.347 ***
	(2.77)	(2.95)	(2.96)	(2.94)
Score of positive friends	4.444 ***	3.826 ***	3.805 ***	3.690 ***
	(6.15)	(5.27)	(5.19)	(3.66)
Classmate relationship	1.799 **	2.073 **	2.071 **	1.469 *
	(2.29)	(2.56)	(2.54)	(1.70)
Cognitive ability	4.872 **	5.162 ***	5.121 ***	6.770 **
	(2.56)	(2.80)	(2.85)	(2.11)
Constant term	133.790 ***	138.369 ***	138.248 ***	44.005 ***
	(6.28)	(9.27)	(9.39)	(2.62)
First-stage regression results of instrumental variables				
Parent’s company of peer (IV1)		0.480	0.418	0.460
Absolute t value		4.86	4.15	6.22
Abundance of individual-participated activities (IV2)			0.234	0.194
Absolute t value			3.47	3.53
F value for first-stage IV		23.598	21.476	24.191
*p* value of Anderson-Rubin test		0.000	0.000	0.000
Sample size	6524	6676	6673	6673
Characteristics and fixed effects at each level	Yes	Yes	Yes	No
Overidentification test		No	No	0.575

Note: t values in brackets, * *p* < 0.10, ** *p* < 0.05, and *** *p* < 0.01. In addition, the control variables of peer support behavior are in accordance with Quan Xiaojuan and Lu Chun (2020) [13] and Wang Fu Bai Hui (2018) [6].

**Table 4 ijerph-20-02480-t004:** Analysis of Intermediary Effect of Sport Atmosphere in Class.

Intermediate Variable	Sports Atmosphere in Class	
	IV1	IV2
Coefficient *a*	1.009 ***	1.007 ***
	(0.03)	0.03
Coefficient *b*	4.892 ***	4.705 ***
	(1.58)	(1.58)
Total effect	23.888 ***	24.240 ***
	(3.52)	(3.43)
Direct effect	13.892 ***	19.502 ***
	(3.87)	(3.78)
Indirect effect	4.936 ***	4.738 **
	(1.60)	(1.60)
Sample size	6687	6698
County/district fixed effect	Yes	Yes
*p* value of Sobel test	0.002	0.003

Note: t values in brackets, ** *p* < 0.05, and *** *p* < 0.01.

## Data Availability

The data are available on request from the corresponding author. All data relevant to the study are included in the article.

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
