# Peer review of "The Peer Effect in Promoting Physical Activity among Adolescents: Evidence from the China Education Panel Survey"

_ijerph, 2023, doi:10.3390/ijerph20032480_

Round 1
Reviewer 1 Report
For studies for children and adolescents, intervention or observational studies may be difficult. However, the introduction of the instrumental variable approach to such a subject can contribute to steady progress. Some sentences, particularly those in the Literature Review section, may be deleted to shorten the manuscript, but most of those are also indicative and make it easier to understand.
There are some concerns as follows:
Please align significant figures throughout the paper.
L.4 and 385
“6.9–8.9” minutes don’t appear in the text or tables, although “8.98” minutes appear in L.239 or “7.34” in L.248.
L.12 and 392
“Health China Action” and “Healthy China Initiative” should be briefly explained with citations.
L.56
In the Introduction section, there is no need to present the main finding and the structure for the paper.
L.68
“Coleman Report” should be cited, also.
L.137, 160, and 170
A reference number should be added to “Ryan (2017)”, which seems Ref.12.
L.160
A reference number should be added to “Angrist (2014)”, which may be different from Ref.27.
L.191
It seems better to move “4.1. Data Source and Descriptive Statistics” except for the last sentence and Table 1 to “3. Study Design”. There are several similar parts in the following sentences.
L.194
I understand that it was a random sampling, but How were 28 county-level units (counties, districts, and cities) and 112 schools and 438 classes selected? For example, was the size of the population taken into account? As a result, are these cohorts representative of the country?
Reviewer 2 Report
First, I wanted to thank you for the opportunity to review your manuscript. It is an interesting topic and I believe that when establishing intervention policies it is necessary to detect all the variables that may influence children's adherence to physical activity. Thus, the mathematical model they propose is interesting insofar as it allows us to isolate the effect in cases where the correlations between the study variables are so strong. However, the manuscript presents a number of weaknesses and there are certain aspects that need to be considered. In general terms, the structure of the article is confusing, does not follow a logical order and jumps from one section to another. The objective of the article should be revised, since in addition to peer influence, it discusses the type of analysis that should be performed to detect such influence. Please take into consideration the suggestions I make below more specifically.
Introduction:
· Line 23: Correct “to e correlation analysis”.
· There are no references in paragraph from line 26 to 35.
· The objective (line 36-37) should be at the end of the introduction.
· Line 37-41 All of this information should be included in the methodology.
· Line 41-46 the hypothesis, should be before the objective.
· Line 47- 61 this paragraph should be included in the discussion.
· The introduction implies a review of the literature. These two sections (introduction and literature review) have to be joined and the information organized concluding in the objective of the article.
Methodology
The purpose of this section is not to explain the mathematical theory from which the applied method derives. They should point out the type of study design, describe the sample (take it from “Data source and descriptive statistics”, the type of sampling, define the variables and finally refer to the statistical method applied with the program used (summarize the information in the current methodology section and integrate lines 231-234).
Results
Some of the information included should be in the methods or discussion.
Discussion
Please organize the discussion as follows:
Discussion of the results obtained and comparison with other studies (here you can include the mathematical models used by other authors, but in a summarized and interpreted form).
Reviewer 3 Report
The topic is interesting, but the presentation is in a poor quality. I suggest the authors find an English editing service for the English and paper formatting.
Round 2
Reviewer 2 Report
I think the manuscript has improved considerably. However, I insist that in order to be published in the journal the objective be moved to the end of the introduction and prior to the methodology (line 99).
